# Terpenoids: Natural Compounds for Non-Alcoholic Fatty Liver Disease (NAFLD) Therapy

**DOI:** 10.3390/molecules28010272

**Published:** 2022-12-29

**Authors:** Pengyu Yao, Yajuan Liu

**Affiliations:** 1Shandong Laboratory of Engineering Technology, Suzhou Institution of Biomedical Engineering and Technology, Chinese Academy of Sciences, Jinan 250101, China; 2Department of Febrile Diseases, Shandong University of Traditional Chinese Medicine, Jinan 250355, China

**Keywords:** non-alcoholic fatty liver disease (NAFLD), natural products, terpenoids, mechanisms, treatment

## Abstract

Natural products have been the most productive source for the development of drugs. Terpenoids are a class of natural active products with a wide range of pharmacological activities and therapeutic effects, which can be used to treat a variety of diseases. Non-alcoholic fatty liver disease (NAFLD), a common metabolic disorder worldwide, results in a health burden and economic problems. A literature search was conducted to obtain information relevant to the treatment of NAFLD with terpenoids using electronic databases, namely PubMed, Web of Science, Science Direct, and Springer, for the period 2011–2021. In total, we found 43 terpenoids used in the treatment of NAFLD. Over a dozen terpenoid compounds of natural origin were classified into five categories according to their structure: monoterpenoids, sesquiterpenoids, diterpenoids, triterpenoids, and tetraterpenoids. We found that terpenoids play a therapeutic role in NAFLD, mainly by regulating lipid metabolism disorder, insulin resistance, oxidative stress, and inflammation. The AMPK, PPARs, Nrf-2, and SIRT 1 pathways are the main targets for terpenoid treatment. Terpenoids are promising drugs and will potentially create more opportunities for the treatment of NAFLD. However, current studies are restricted to animal and cell experiments, with a lack of clinical research and systematic structure–activity relationship (SAR) studies. In the future, we should further enrich the research on the mechanism of terpenoids, and carry out SAR studies and clinical research, which will increase the likelihood of breakthrough insights in the field.

## 1. Introduction

In the epidemiology of liver disease, there has been a gradual transition in focus from infectious diseases to metabolic diseases. Non-alcoholic fatty liver disease (NAFLD) has become a serious public health issue, affecting the health of approximately one-quarter of adults worldwide, causing wide-ranging social and economic implications [1]. The prevalence of NAFLD is 25% globally, and it has become the most rapidly increasing cause of liver-related mortality [2]. The prevalence of NAFLD is increasing rapidly worldwide and is predicted to become more prevalent in the future as the obese and diabetic populations increase [3].

NAFLD is a clinicopathological syndrome characterized by parenchymal cell steatosis and fat storage without history of excessive alcohol consumption. Its disease pathophysiological development ranges from non-alcoholic hepatic steatosis to non-alcoholic steatohepatitis (NASH) and hepatic fibrosis, potentially evolving into hepatic cirrhosis, hepatocellular carcinoma (HCC), and liver failure. The prevalence of NASH is approximately 30% for patients with NAFLD, and approximately 20% of NASH patients with fibrosis progress to cirrhosis [4]. However, NAFLD is a diagnosis of exclusion rather than one of inclusion, and so the new nomenclature “metabolic-associated fatty liver disease (MAFLD)” has been proposed [5]. The standardization of the nomenclature for NAFLD still needs to be explored. So far, the above two terms have been accepted. NAFLD is a complex disease in which disease severity is influenced by genetic, environmental, and behavioral factors (e.g., diet, physical activity, and socioeconomic influences) [6]. NAFLD is considered the hepatic manifestation of metabolic syndrome and involves pathological changes in multiple systems, including metabolic disease (such as metabolic syndrome, type 2 diabetes mellitus, obesity, and hypertension) [7,8], cardiovascular disease (such as coronary heart disease, atherosclerosis, cardiomyopathy, and arrhythmia) [9], extrahepatic carcinomas (such as incident gastric and colorectal cancer) [10], and chronic kidney disease [11].

The underlying pathophysiological mechanism of NAFLD remains unclear, although it is typically characterized by an accumulation of lipids in the liver that stems from multiple factors. Furthermore, it coexists with inflammation, hepatic cell injury, and the deposition of collagen fibers. The “two-hit hypothesis” explains how simple fatty liver or steatosis progresses to severe NASH and liver fibrosis [12], in the pathogenesis of early NAFLD. With the in-depth study of the pathological mechanism, this view seems unable to fully summarize the complexity of NAFLD, and the “multiple impact model” has been proposed and widely used. Factors such as lipid accumulation, insulin resistance [13], oxidative stress [14], the gut microbiome [15], and inflammation [16] have been implicated in the pathogenesis of NAFLD, but the mechanisms that drive disease progression are not fully understood. NAFLD is a disease with a favorable prognosis, but when NAFLD progresses to NASH, it can be fatal, as inflammation and fibrosis progress to end-stage liver disease. Therefore, the prevention and early treatment of NAFLD are very important. Despite this high prevalence of NAFLD, there are currently no approved treatments.

The identification of drug candidates that can successfully treat NAFLD, with few or no side effects, is a huge challenge. Natural products have played an important role in drug discovery and were the basis of most early medicines [17], especially in the treatment of chronic and metabolic diseases. Some of the most famous drugs worldwide, including aspirin, morphine, artemisinin, berberine, and paclitaxel, are derived from natural sources [18]. With the advances in natural medicinal chemistry technology, it is possible to determine the chemical composition of plants and their application in drug discovery. Over the nearly four decades from 1981 to 2019, natural products, as sources of new drugs, still exist and account for a large share of new drug discovery [19]. Terpenoids are a class of active natural products with a wide range of pharmacological effects and that represent a rich reservoir of candidate compounds for drug discovery [20]. Based on their chemical structure, terpenoids are composed of several subclasses, including hemiterpenoids, monoterpenoids, sesquiterpenoids, diterpenoids, sesterterpenoids, triterpenoids, and tetraterpenoids. Terpenoids have been widely used for the treatment of many diseases due to their broad range of biological activities, such as anti-microbial, anti-cancer, hypotensive, anti-hyperlipidemic, anti-hyperglycemic, anti-inflammatory, anti-oxidant, anti-parasitic, immunomodulatory, and anti-cholinesterasic activities [21]. The therapeutic effects of terpenoids in NAFLD have been increasingly discussed, suggesting their great potential in the treatment of NAFLD, and potentially representing a new direction for breakthroughs in drug development. There are many studies on terpenoids in NAFLD, but there is still a lack of a systematic analysis of these therapeutic applications. The purpose of this article is to review the recent research progress concerning terpenoids in the treatment of NAFLD and the underlying mechanism of action, and to provide a comprehensive introduction of this class of compounds and their potential in the treatment of NAFLD.

## 2. Methods

We searched the electronic databases PubMed, Excerpt Medica, Web of Science, Science Direct, and Springer for the period 2011–2021 regarding the use of terpenoids to treat NAFLD, using the following search terms: (“terpenoid” OR “monoterpenoids” OR “sesquiterpenoids” OR “ diterpenoids” OR “triterpenoids” OR “tetraterpenoids”) AND (“non-alcoholic fatty liver disease” OR “metabolically associated fatty liver disease” OR “non-alcoholic fatty liver” OR “non-alcoholic steatohepatitis ” OR “NAFLD” OR “NAFL” OR “MAFLD” OR “NASH”).

Studies were excluded from this review if they were found to harbor significant methodological errors or to lack scientific value. In order to aid in classification efforts, studies regarding mixtures of different compounds or crude extracts were also excluded from this study, in addition to those focused on terpenoids with poorly defined chemical structures. Not all terpenoids have been described in detail in the literature; for example, studies of the carotenoid family often ignore their presence as terpenoids. Therefore, according to the results of the literature search, we conducted a secondary literature search for the obtained terpenoids. The limitations of the first search were supplemented and our research system was greatly enriched. Finally, 43 terpenoids were obtained.

In total, 43 natural compounds were classified into five categories according to their structure, namely monoterpenoids, sesquiterpenoids, diterpenoids, triterpenoids, and tetraterpenoids. Figure 1 shows the numbers of the different types of terpenoids.

## 3. Terpenoids and Their Mechanism of Action in the Treatment of Non-Alcoholic Fatty Liver Disease

Increasing evidence indicates that terpenoids can effectively inhibit the progression of FALD, play a therapeutic role in different stages of the pathological process of disease, and effectively prevent and treat FALD through a variety of approaches, including improving lipid metabolism, inhibiting oxidative stress, inhibiting inflammation, and preventing fibrosis. Table 1 provides and introduction to the basic information of terpenoids, including the subclass, compounds, molecular formula, weight, and sources. Table 2 summarizes the effects and the mechanisms of action of 43 terpenoids, including the animal/cell model, dosage, target, mechanism, and effect. Figure 2 shows the chemical structure of 43 terpenoids. We will classify these terpenoids according to their chemical structure and introduce the results of research into their use in the treatment of NAFLD.

### 3.1. Monoterpenoids

Monoterpenoids are compounds with a basic carbon frame that consists of two isoprene and C10, constituting a large family of natural products. Iridoids belong to the monoterpenoids. In total, our study identified nine monoterpenoids capable of treating NAFLD.

#### 3.1.1. Paeoniflorin

Paeoniflorin is a monoterpenoid glycoside extracted from *Paeonia albiflora* Pallas, which has been extensively studied due to its remarkable pharmacological effects, low toxicity, and few side effects [22]. Paeoniflorin has been reported to possess anti-inflammatory, anti-oxidative stress, anti-hyperlipidemic, and hepatoprotective activities [23]. The anti-NAFLD effect of paeoniflorin is involved in many aspects. It could be used to improve lipid metabolism, improve glucose metabolism, and inhibit inflammation for the protection of liver function [24]. Insulin is a hormone with multiple functions, among which is to inhibit the lipolysis of adipose tissue. The first hit of the “two-hit hypothesis” is that insulin resistance causes hepatic steatosis. The insulin-sensitizing effect of paeoniflorin is among the reasons behind its efficacy in preventing NAFLD. Paeoniflorin could exert an insulin-sensitizing effect by regulating the insulin signaling pathway IRS/Akt/GSK3β [25]. This is also why paeoniflorin can be of benefit for a variety of metabolic diseases, such as diabetes, hypertension, hyperlipidemia, and NAFLD. The activation of the LKB1/AMPK signaling pathway is another key mechanism by which paeoniflorin exerts its effects. The protective effects of paeoniflorin might be involved in the activation of LKB1/AMPK signaling pathways, which results in the inhibition of lipogenesis, insulin resistance, and hepatic steatosis [26]. NAFLD is a chronic inflammatory disease, and anti-inflammatory therapy is beneficial throughout its pathological process. The significant anti-inflammatory effect of paeoniflorin is key in preventing NAFLD from transforming into NASH, and the inhibition of the ROCK/NF-κB signaling pathway in the NASH liver is associated with the mechanism of its anti-inflammatory action [27].

#### 3.1.2. Catalpol

Catalpol is an iridoid compound widely distributed in many plant families and is mainly extracted from the roots of *Rehmannia glutinosa* (Gaertn.) Libosch. ex Fisch. and C. A. Mey. Catalpol possesses extensive pharmacological activities, such as anti-inflammatory, anti-oxidative stress, anti-diabetic, anti-stroke, and anti-osteoporosis activity [28]. Catalpol’s role in metabolic diseases is widely recognized. Activated AMPK can enhance lipocatabolism and produce ATP to regulate lipid metabolism. Catalpol regulates lipid metabolism through the activation of AMPK, thereby reducing hepatic steatosis [29]. AMPK is a potential target for treating diseases, which is often associated with changes in metabolism [30]. Catalpol induces autophagy and attenuates lipotoxicity, the mechanism of action of which is also through AMPK activation [31].

The activation of the p66shc/cytochrome C cascade is responsible for causing ROS metabolism, hepatic steatosis, and apoptosis in NAFLD. The anti-oxidative stress and liver-protective effects of catalpol are dependent on the miR-96-5p/p66shc/cytochrome C cascade [32]. In addition, anti-inflammatory activity has been shown to be a key effect of catalpaol in the treatment of NAFLD [33]. Due to the complexity of the mechanism of action, the underlying mechanism of catalpol’s anti-inflammatory effect has not been fully elucidated.

#### 3.1.3. Geniposide

Geniposide, as a kind of iridoid glycoside extracted from *Gardenia jasminoides* J. Ellis, has many biological effects, such as anti-inflammatory, hepatoprotective, and cholagogic effects [34]. Insulin resistance leads to fat accumulation in the liver, which is the first hit in the onset of NAFLD. Excess fat accumulation in the liver leads to a vicious cycle of aggravated insulin resistance and induced oxidative stress, which increases the pathological complexity of NAFLD. Geniposide inhibits lipid accumulation via enhancing anti-oxidative stress and anti-inflammation activity, which mostly depend on upregulating the Nrf2/HO-1 and AMPK signaling pathways [35]. Geniposide exerts protective effects against hepatic steatosis, the underlying mechanism of which may be associated with its regulation of adipocytokine release and expression of PPARa [36].

#### 3.1.4. Genipin

Genipin is a metabolite derived from genipioside and has been recognized as a beneficial compound against metabolic disorders. For decades, genipin has been extensively studied and used in the field of liver disease, including in the treatment of acute liver injury, fulminant hepatitis, NAFLD, and other non-cancer liver diseases [37]. Genipin potentially serves as an effective therapeutic intervention against NAFLD, effectively antagonizing hyperlipidemia and hepatic lipid accumulation, by regulating the miR-142a-5p/SREBP-1c axis [38]. Pyroptosis not only causes liver cell death, but also aggravates the inflammatory response and process of fibrosis. Pyroptosis plays an important role in the development of NAFLD; thus, it is important to stop pyroptosis to prevent disease. Genipin reverses liver damage and inhibits UCP2-mediated pyroptosis [39].

#### 3.1.5. Sweroside

Sweroside is a natural product found in *Swertia bimaculata* (Siebold and Zucc.) Hook. f. and Thomson ex C. B. Clarke. Sweroside is a typical iridoid that exhibits diverse biological activities, such as hepatoprotective, anti-diabetic, and anti-inflammatory effects [40]. The peroxisome proliferator-activated receptors (PPAR-α, PPAR-β/δ, and PPAR-γ) are members of the nuclear receptor super-family and play crucial roles in glucose and lipid metabolism [41]. The activation of PPARα induces the oxidation of fatty acids, and regulates liver fat metabolism. Sweroside may ameliorate obesity and the fatty liver via the regulation of lipid metabolism and its anti-inflammatory activity; these effects are closely associated with the regulation of PPAR-α [42]. The activation of the NLRP3 inflammasome increases NASH, induces intense inflammatory responses, causes pyrodeath, increases lipid accumulation, and promotes fibrosis. Sweroside can inhibit these pathological changes by blocking the activation of the NLRP3 inflammasome in macrophages and liver tissues [43].

#### 3.1.6. Swertiamarin

Swertiamarin is a typical natural iridoid found in *Swertia bimaculata* (Siebold and Zucc.) Hook. f. and Thomson ex C. B. Clarke, which has been reported to cure many metabolic diseases, such as diabetes and hyperlipemia [44]. Hepatic steatosis is a common liver pathological lesion, defined as the presence of large and small vesicles of fat, predominantly triglycerides, accumulating within hepatocytes. Swertiamarin ameliorates obesity and improves insulin resistance by improving dyslipidemia and attenuating inflammation, which further improves hepatic steatosis. The p38 MAPK and NF-κB pathways participate in some of its effects, but its mechanism of action is still not fully understood [45].

#### 3.1.7. Aucubin

Aucubin is widely distributed in plants, such as Eucommia ulmoides Oliv., Plantago asiatica L., and Scrophularia ningpoensis Hemsl. It has shown many positive effects, such as anti-oxidant, anti-aging, anti-inflammatory, anti-fibrotic, anti-cancer, hepatoprotective, neuroprotective, and osteoprotective properties [46]. Nrf2 is the main regulator of the cellular defense system against oxidative stress. Nrf2, similar to AMPK, is often concerned in the treatment of NAFLD. Aucubin inhibits lipid accumulation, the inflammatory response, and oxidative stress via the Nrf2/HO-1 and AMPK signaling pathways [47].

#### 3.1.8. Gentiopicroside

Gentiopicroside, the main active ingredient of *Gentiana scabra* Bunge, has anti-inflammatory, anti-fibrosis, anti-oxidative stress, and anti-apoptosis activity [48]. These effects make gentiopicroside a significant contributor to the treatment of NAFLD. Gentiopicroside may be a useful therapeutic strategy for NAFLD through the alleviation of oxidative damage and lipid accumulation in the liver. The upregulation of the Nrf2 anti-oxidant pathway is thought to be a key mechanism behind its function [49].

#### 3.1.9. Geraniol

Geraniol is an acyclic isoprenoid monoterpene isolated from the essential oils of aromatic plants, including *Elsholtzia ciliata* (Thunb.) Hyl., *Murraya exotica* L., *Rosa rugosa* Thunb., and several other plants. Treatment with geraniol reduced blood lipids, attenuated hepatic fibrosis and apoptosis, and suppressed inflammation in NASH [50]. The mechanism of action of geraniol in NAFLD has not been thoroughly studied, and its pathway and target in this disease have not been mentioned in the literature.

### 3.2. Sesquiterpenoids

Sesquiterpenoids are a class of enormously diverse natural products derived from a 15-carbon precursor, contains three isoprene units. Sesquiterpenoids have a variety of skeletal structures due to the diversity of sesquiterpene hydrocarbon backbones. In total, our study identified three sesquiterpenoids capable of treating NAFLD.

#### 3.2.1. Curcumol

Curcumol, as an important component of *Curcuma phaeocaulis* Valeton, structurally belongs to the guaiacane sesquiterpene natural products. While curcuminoids have been extensively studied for their anti-microbial, anti-oxidant, anti-inflammatory, and other effects, the therapeutic efficacy of curcumol is still emerging [51]. The treatment of NAFLD using curcumol is a research hotspot. Cellular senescence has attracted much interest from researchers due to its involvement in NAFLD. Hepatocyte senescence can be found in the liver of NAFLD patients, which aggravates steatosis in the liver. When the liver is injured, the ferritinophagy signaling pathway can be activated, inducing the decomposition of ferritin storage and iron-rich mitochondrial proteins, releasing excess iron into the cytoplasm, leading to disordered iron metabolism in the liver, and eventually pathological iron overload. Excess iron can cause cell senescence, and further promote the pathological progression of NAFLD. Previous research has clarified the mechanism of the curcumol inhibition of hepatocyte senescence through the YAP/NCOA4 regulation of ferritinophagy in NAFLD [52]. In addition, curcumerol has a significant protective effect on liver function and liver fibrosis, which may be related to the regulation of the TLR4, TAK1, and NF-κB/p65 signaling pathways to reduce inflammatory factors and increase anti-inflammatory factors [53].

#### 3.2.2. β-Patchoulene

Patchoulene is among the natural compounds derived from *Pogostemon cablin* (Blanco) Benth. In the volatile oil of patchouli there exist four isoforms, of which β-patchoulene is one. β-patchoulene is well known for its anti-inflammatory and anti-oxidative functions in various diseases. β-patchoulene exerts a positive effect against NASH by interrupting the vicious cycle between oxidative stress, histanoxia, and lipid accumulation. The activation of the CD36/AMPK signaling pathway to balance lipid metabolism disorders is its internal mechanism [54,55].

#### 3.2.3. β-Caryophyllene

β-caryophyllene is an odoriferous bicyclic sesquiterpene found in various herbs and spices, such as *Rosmarinus officinalis* L., *Cinnamomum cassia* (L.) D. Don, *Ocimum basilicum* L., and *Lavandula angustifolia* Mill. β-caryophyllene has potential efficacy in preventing and ameliorating non-alcoholic fatty liver disease and its associated metabolic disorders. AMPK is considered to be an effective target for regulating insulin synergism in the treatment of metabolic diseases, and has been highly studied in NAFLD. β-caryophyllene could attenuate lipogenesis and lipid accumulation by upregulating the AMPK signaling pathway. Further mechanistic studies revealed that the β-caryophyllene-induced activation of AMPK could be mediated by the CB2 receptor-dependent Ca2^+^/CaMKK signaling pathway [56].

### 3.3. Diterpenoids

Diterpenoids are terpenoids composed of four isoprene structural units. More than 126 different diterpenoid carbon skeletons have been identified, which can give rise to more than 18,000 compounds [57]. Diterpenoids are widely exploited in the clinic and in research. In total, our study identified five diterpenoids capable of treating NAFLD.

#### 3.3.1. Carnosic Acid

Carnosic acid is a phenolic diterpene isolated from *Rosmarinus officinalis* Linnaeus and *Salvia japonica* Thunb., which possesses anti-tumor, anti-inflammatory, neuroprotection, anti-oxidative, and anti-microbial properties [58,59]. The effectiveness of carnosic acid in preventing fat accumulation, especially in fatty livers, and in relieving glucose intolerance has been proven [60]. Studies have shown that carnosic acid is an effective anti-obesity agent that regulates fatty acid metabolism in C57BL/6J-ob/ob mice. It can regulate the expression of hepatic lipogenesis-related genes (L-FABP, SCD1, and FAS), which decreased, whereas lipolysis-related gene (CPT1) expression increased [61]. However, other studies have confirmed that carnosic acid has a certain level of hepatotoxicity in a dose-dependent manner, so a proper safety assessment is required before use [62]. How to avoid the toxic damage of drugs and give full play to their effects is a problem worth discussing.

Carnosic acid may be used as a potential therapeutic agent in the treatment of NAFLD-related metabolic diseases. Related research has revealed that carnosic acid possesses the ability to improve high-fat-diet-induced NAFLD in mice through reducing the lipogenesis and inflammation in the liver [63]. Studies have found a link between liver cell apoptosis and the miR-34a/SIRT1/p66shc pathway, which can be regulated by CA in NAFLD [64]. Brain damage in non-alcoholic fatty liver disease is clearly present and has been demonstrated in many studies. Data in the literature suggest a role of NAFLD in promoting early cerebral alterations with cognitive impairment, subclinical ischemic lesions, and cerebrovascular accidents [65]. HFD-induced NAFLD can exacerbate dopaminergic neuron injury, while carnosic acid is able to prevent such impairment [66]. The development of this compound may provide a new path for the prevention and treatment of NAFLD complications.

#### 3.3.2. Ginkgolide (A,B)

Ginkgolide, a natural substance extracted from *Ginkgo biloba* L., is a terpenoid compound composed of sesquiterpene lactone and diterpene lactone. Eleven terpene lactones were isolated from ginkgo biloba leaves. Ginkgolides A, B, C, J, K, L, M, N, P and Q belong to the diterpene lactones, while ginkgolides belong to the sesquiterpene lactones [67]. Ginkgolide has been widely used in clinical practice because of its rich pharmacological effects and weak side effects. It has anti-oxidation, anti-inflammation, anti-aging, anti-platelet aggregation, and anti-apoptosis effects, and lowers blood pressure, promotes blood circulation, and protects the central nervous system.

Ginkgolide A may be a natural compound with great therapeutic promise, especially for the treatment of cardiovascular, hepatological, and neurological diseases [68]. Ginkgolide A is non-toxic at high concentrations, and may be feasible as a therapeutic agent for NAFLD patients. Studies have confirmed that ginkgolide A’s effects on NAFLD mainly include anti-oxidative stress and anti-inflammation activity. Ginkgolide A showed hepatoprotective efficacy by inducing cellular lipoapoptosis and by inhibiting inflammation [69].

Ginkgolide B is a diterpenoid compound isolated from ginkgo biloba leaves, which is the most significant active component in active lactones. Pregnane X receptor (PXR, NR1I2) is a ligand-activated nuclear hormone receptor and its value in metabolic diseases has been emphasized in recent years. Targeting PXR may be a strategy for the therapy of metabolic diseases. Ginkgolide B may have beneficial effects on metabolic disorders, possibly through the activation of PXR, such as blocking body weight gain, attenuating hypertriglyceridemia and hepatic steatosis, and improving bile acid homeostasis in DIO mice [70].

The ferroptosis of hepatocytes and intrahepatic macrophages may lead to the progression of simple fatty liver degeneration to NASH, and the inhibition of ferroptosis is gradually becoming a new treatment strategy for NAFLD. Ginkgolide B treatment has a specific effect on lipid accumulation and oxidative-stress-induced ferroptosis in NAFLD, the mechanism of action of which is through the regulation of the Nrf2 signaling pathway [71].

#### 3.3.3. Acanthoic Acid

Acanthoic acid is a pimaradiene diterpene isolated from the root of *Eleutherococcus senticosus* (Rupr. and Maxim.) Maxim. The treatment of liver disease is an important aspect of acanthoic acid applications, and the value of acanthoic acid in liver disease has been widely explored, having been studied in alcoholic liver disease [72], drug-induced hepatotoxicity [73], and fulminant liver failure [74]. Acanthoic acid may be proved to be an attractive candidate for the treatment of NAFLD, as it can attenuate liver steatosis and fibrosis in NAFLD. The farnesoid X receptor (FXR) and liver X receptor (LXR) are involved in lipid metabolism, glucose metabolism, and inflammatory activation, and maintain the nutrient/energy balance of the liver. Acanthoic acid can regulate fat metabolism, and especially prevent lipid accumulation and fatty acid synthesis, the mechanism of which is to activate the FXR and LXR signaling pathways, contributing to the increased expression of the AMPK-SIRT1 signaling pathway [75].

#### 3.3.4. Dehydroabietic Acid

Dehydroabietic acid is a diterpene found in tree pine, derived from Pinaceae plants such as *Pinus massoniana* Lamb. and *Picea asperata* Mast. Various bioactive effects of dehydroabietic acid have been studied, including anti-bacterial, anti-fungal, and anti-cancer activities [76]. The accumulation of iron-dependent lipid peroxides is among the important causes of NAFLD. Dehydroabietic acid can improve NAFLD by regulating lipid metabolism and inhibiting ferroptosis, and its ability to reduce triglyceride (TG), total cholesterol (TC), and lipid peroxidation levels is significant [77].

### 3.4. Triterpenoids

Triterpenoids, composed of four isoprene structural units, are derived from (E, E, E) -geranylgeranyl diphosphate (GGPP). More than 126 different triterpenoid carbon skeletons have been identified, which give rise to more than 18,000 compounds [78]. The complexity and diversity of the structure of triterpenoids results in different biological activities, and are widely exploited in the clinic and in research. In total, our study identified 20 triterpenoids capable of treating NAFLD.

#### 3.4.1. Ginsenoside (Rg1, Rg2, Rh1, Rb1, Rb2 and Mc1)

*Panax ginseng* C. A. Mey. is among the most widely used natural medicinal plants worldwide. Ginsenosides, the major bioactive constituents of *Panax ginseng* C. A. Mey., are a series of glycosylated triterpenoids which belong to protopanaxadiol (PPD)-, protopanaxatriol (PPT)-, ocotillol (OCT)-, and oleanane (OA)-type saponins. Approximately 300 ginsenosides have been isolated and identified from different Panax species [79]. The known ginsenoside monomers can be divided into three categories: ginsenodiols (such as Rb1, Rb2, Rc, Rd, and F2), ginsenotriols (such as Re, Rg1, Rg2, Rf, and Rh1), and pentacyclic triterpenoid saponins (such as RO). Ginsenoside Rb1, Rb2, Rg1, Rg2, Rh1 and Mc1 and other components have been proven to have liver-protective effects and can be used to treat NAFLD.

Ginsenoside Rg1, a bioactive phytochemical, is the most reported ginsenoside in the field of NAFLD therapeutic research. Improving fat synthesis and metabolism is an important measure of its function. Ginsenoside Rg1 significantly improves fat metabolism and synthesis, inhibits lipid synthesis, decreases lipid uptake, enhances lipid oxidation and reduces hepatic steatosis, by regulating PPAR ɑ and PPAR γ expression [80]. FOXO1 may be involved in many aspects of liver pathology, such as hepatic aging, steatosis, and glucose and lipid metabolism dysregulation. Ginsenoside Rg1 exerts the pharmaceutic effect of maintaining FOXO1 activity in the liver, thus protecting livers from senescence- and metabolic-abnormality-induced fatty liver disease [81]. Ginsenoside Rg1 may protect NAFLD through inflammation; the anti-inflammatory activity of ginsenoside Rg1 includes the inhibition of endoplasmic reticulum(ER) stress and inflammasome activation [82]. The changes in the transcriptome also suggest that the efficiency of ginsenoside Rg1 treatment on NAFLD may be associated with two hub genes, Atf3 and Acox2 [83]. This has been demonstrated in its ability to improve liver function and alleviate pathological processes in animal models of NAFLD.

Ginsenosides Rg2 and Rh1 belong to the category of ginsenotriols. Saponin extract contains amounts of ginsenosides Rg2 and Rh1, which can inhibit inflammation-mediated pathological inflammasome activation in macrophages, thereby preventing NAFLD development [84]. In addition to its anti-inflammatory effects, the administration of ginsenosides Rg2 significantly ameliorated HFD-induced hepatic oxidative stress and apoptosis in a SIRT1-dependent manner [85]. The effects of these compounds are not singular, but complex and multifaceted. Ginsenoside Rh1 has a positive effect on NAFLD via its anti-fibrotic and hepatoprotective activity [86].

Ginsenosides Rb1 and Rb2 belong to the category of ginsenodiols. Ginsenoside Rb1 can reduce liver cell apoptosis, and the activation of PPAR-γ may be the internal mechanism [87]. Ginsenoside Rb2 has a significant positive effect on NAFLD and glucose intolerance, and the underlying molecular mechanism consists of alleviating hepatic lipid accumulation and restoring hepatic autophagy via sirt1 induction and AMPK activation [88].

Ginsenoside Mc1, a newly identified deglycosylated ginsenoside, is converted from the major ginsenoside Rc. Ginsenoside Mc1 exerts protective effects against apoptotic damage, insulin resistance and lipogenesis in the liver [89].

Therefore, ginsenoside supplementation could be a potential therapeutic strategy to prevent NAFLD in patients. Of course, the therapeutic value of many different ginsenosides in NAFLD remains to be explored.

#### 3.4.2. Ursolic Acid

Ursolic acid is a natural triterpene compound and is widely distributed in nature. It is known to be found in at least hundreds of plants, many of which are aromatic plants and fruit trees used in traditional Chinese medicine. Ursolic acid exists in various forms in plants, some in the free state, some in the binding state of ester or glycoside, and some as other polysubstituted derivatives. Ursolic acid has widespread pharmacologic activity, including anti-tumor, anti-inflammatory, anti-oxidant, anti-apoptotic, anti-allergy, and anti-carcinogenic effects [90]. Ursolic acid effectively ameliorated HFD-induced hepatic steatosis through a PPAR-an involved pathway, via improving key enzymes in the control of lipid metabolism [91]. Ursolic acid treatment significantly prevents the development of NAFLD and liver injury in db/db mice, most likely through increasing lipid β-oxidation and inhibiting hepatic ER stress [92]. The LXRα is a multifunctional nuclear receptor that controls lipid homeostasis. Ursolic acid, a novel LXRα antagonist, can inhibit adipogenesis through this mechanism in NAFLD [93]. Ursolic acid administration not only has therapeutic effects, but can also prevent the occurrence of disease when ingested at an early stage. The administration of ursolic acid during periods of developmental plasticity shows prophylactic potential against dietary-fructose-induced NAFLD [94].

#### 3.4.3. Betulinic Acid

Betulinic acid is a pentacyclic triterpene distributed in a variety of plants, such as *Betula platyphylla* Sukaczev. It shows a wide spectrum of biological and pharmacological properties, such as anti-inflammatory, anti-diabetic, and anti-hyperlipidemic effects. The FXR plays an important role in hepatic homeostasis. The activation of FXR can reduce lipotoxicity and increase cholesterol excretion, thus playing a role in improving insulin resistance, with anti-inflammatory and anti-fibrosis effects. Betulinic acid is a novel FXR agonist, and can alleviate endoplasmic reticulum stress-mediated NAFLD through the activation of FXR. The effect of BA is via the FXR-mediated inhibition of PERK/EIF2α/ATF4/CHOP signaling [95]. Betulinic acid can regulate fat metabolism by delaying lipid accumulation, reducing fat synthesis, inhibiting hepatic steatosis, and promoting fatty acid oxidation, among other effects. Betulinic acid effectively ameliorates intracellular lipid accumulation in liver cells by modulating the AMPK-SREBP signaling pathway [96]. Betulinic acid protects hepatocytes from abnormal lipid deposition in NAFLD through the YY1/FAS pathway [97].

#### 3.4.4. Glycyrrhizic Acid

Glycyrrhizic acid is a triterpene glycoside isolated from *Glycyrrhiza uralensis* Fisch., which is an edible and medicinal plant. Glycyrrhizin has anti-inflammatory, anti-allergy, and anti-oxidative stress effects. As a sweetener, glycyrrhizin is widely used in all kinds of food. Glycyrrhizin can reduce liver lipogenesis, increase lipid metabolism, reduce liver inflammation, and prevent anti-liver fibrosis by intervening in the pathological process of NAFLD [98]. Glycyrrhizic acid protects against NAFLD through the suppression of lipid synthesis, promoting fatty acid β-oxidation and triglyceride-rich lipoproteins lipolysis, inhibiting gluconeogenesis and the promotion of glycogen synthesis, and reversing insulin resistance [99]. However, the internal mechanism of glycyrrhizin intervention in the pathology of NAFLD remains to be further clarified. The bile acids (BAs) modulate lipid and glucose metabolism, inflammation, and fibrosis, and are closely related to the pathological process of NAFLD [100]. Glycyrrhizic acid alleviates non-alcoholic steatohepatitis via modulating bile acids [101].

#### 3.4.5. Glycyrrhetinic Acid

Glycyrrhetinic acid, a pentacyclic triterpenoid saponin metabolite of glycyrrhizin, is extracted from *Glycyrrhiza uralensis* Fisch. 18β glycyrrhetinic acid, uralenic acid, and enoxolone are the aliases of glycyrrhetinic acid. Vitamin A (VA) plays critical roles in various physiological functions, including the regulation of glucose and lipid homeostasis in the liver [102]. Glycyrrhetinic acid, as a novel AKR1B10 inhibitor, could promote retinoic acid synthesis. Glycyrrhizic acid restored the balance of VA metabolism in NAFLD/NASH mice by metabolizing to glycyrrhetinic acid [103].

Hepatocyte nuclear factor 4-alpha (HNF4α) is best known for its role as a master regulator of liver-specific gene expression. Glycyrrhetinic acid was reported to act as a partial HNF4α antagonist to modulate hepatic very low-density lipoprotein (VLDL) secretion and gluconeogenesis [104]. Free fatty acid (FFA)-induced lipotoxicity plays a crucial role in the pathogenesis of NAFLD. Glycyrrhetinic acid reduces hepatic lipotoxicity by stabilizing the integrity of lysosomes and mitochondria and inhibiting cathepsin B expression and enzyme activity [105].

#### 3.4.6. Oleanolic Acid

Oleanolic acid is a pentacyclic triterpenoid found in many plants, which has been isolated from more than 1620 plant species [106]. It has attracted the attention of the scientific community because of its wide range of biological activities. The regulation of metabolism by oleanolic acid is a combination of its multiple mechanisms of action in NAFLD. The interplay between the gut and liver, the so-called “gut–liver axis” (GLA), has been widely considered as a potential therapeutic target for NAFLD. Oleanolic acid has a protective influence on imbalances in GLA homeostasis, has anti-oxidative stress and anti-inflammatory effects, and improves glucose tolerance and insulin resistance by regulating GLA in NAFLD [107]. The expression of LXRs is correlated with the degree of hepatic fat deposition, as well as with hepatic inflammation and fibrosis in NAFLD patients. Oleanolic acid may be a novel antagonist of LXRα- and PXR-mediated lipogenesis, and plays a role in the treatment of NAFLD by inhibiting liver X receptor alpha and pregnane X receptor to attenuate ligand-induced lipogenesis [108]. In addition, it has been reported that dietary supplementation with oleanolic acid in the neonatal phase of development potentially encourages hepatoprotection against the development of NAFLD in adult life [109].This suggests that oleanolic acid can be used as an early preventative measure. As a derivative of oleanolic acid, 3-acetyl-oleanolic acid shows good potential against NAFLD, including the amelioration of lipid accumulation, anti-steatotic effects, and the inhibition of apoptosis. The therapeutic effect of 3-acetyl-oleanolic may lie in two aspects: altering the secretion of multiple adipokines and activating AMPK-related pathways [110]. Ha-20 is a newly discovered oleanolic acid derivative, which effectively suppresses fat accumulation in the liver. FABP4/aP2 is among the key carrier proteins in fat accumulation, and HA-20 inhibits adipogenesis in a manner involving the PPARγ-FABP4/aP2 pathway [111].

#### 3.4.7. Astragaloside IV

Astragaloside IV, among the major compounds from the extract of Astragalus membranaceus, is a cycloartane-type triterpene glycoside chemical, which has a series of pharmacological effects including anti-inflammatory and anti-oxidant activity and the regulation of energy metabolism [112]. AS-IV attenuates free fatty acid (FFA)-induced ER stress and lipid accumulation via the AMPK signaling pathway, which further supports its use as a promising therapeutic for the treatment of NAFLD [113]. Insulin resistance participates in liver fat accumulation, hepatocyte steatosis, and steatohepatitis, and is the central link between the occurrence and development of NAFLD. AS-IV inhibits PTP1B and effectively improves insulin resistance, and also has an effect on the prevention and treatment of NAFLD [114].

#### 3.4.8. Mogroside V

Mogroside, the main sweet component of *Siraitia grosvenorii* (Swingle) C. Jeffrey ex A. M. Lu and Zhi Y. Zhang, is a triterpene glucoside, and more than 20 triterpene saponins have been isolated and identified. It has the functions of lowering blood glucose, lowering blood lipids, and scavenging free radicals, alongside anti-inflammatory and anti-oxidant effects [115]. Mogrosides protect against the development of NAFLD, and the mechanisms of action underlying this inhibitory effect may be associated with the promotion of AMPK phosphorylation and the enhancement of anti-oxidative defenses through the upregulation of p62 expression [116]. Mogroside V is the representative active ingredient in mogroside, and it has been confirmed to play an important role in NAFLD after in-depth study. Mogroside V exerts a pronounced effect, improving hepatic steatosis and alleviating lipid accumulation through the regulation of the disequilibrium of lipid metabolism in the liver via an AMPK-dependent pathway [117].

#### 3.4.9. Asiatic Acid

Asiatic acid, an active substance of *Centella asiatica* (L.) Urb., belongs to the category of triterpenoids. It possesses a wide spectrum of biological activities, notably anti-inflammatory, anti-diabetic, anti-oxidant, hepatoprotective, and anti-viral effects (specifically, hepatitis C virus) [118]. The overexpression of NF-KB plays an important role in the occurrence and development of liver inflammation, hepatocyte apoptosis, and hepatic fibrosis. Asiatic acid can relieve oxidative stress and inflammation by inhibiting the NF-kB signaling pathway, effectively alleviating hepatic steatosis, hepatocyte apoptosis, and hepatocyte injury [119].

#### 3.4.10. Corosolic Acid

Corosolic acid is a natural pentacyclic triterpenoid, mainly derived from plants such as *Eriobotrya japonica* (Thunb.) Lindl., *Eriobotrya japonica* (Thunb.) Lindl., Actinidia chinensis Planch., and *Lagerstroemia speciosa* (L.) Pers. It exerts anti-diabetic, anti-obesity, anti-inflammatory, anti-hyperlipidemic, and anti-viral effects [120]. Among the many cytokines, TGF-β1 plays the most important role in liver fibrosis. The Smad protein family is involved in TGF-β signa`ling, of which Smad2 is a member of the receptor-activated Smad class. Corosolic acid prevents non-alcoholic liver injury and fibrosis in the progression of NASH through regulating the TGF-β1/Smad2, NF-κB, and AMPK signaling pathways [121].

#### 3.4.11. Arjunolic Acid

Arjunolic acid is a pentacyclic triterpenoid found in *Cymbidium goeringii* (Rchb. f.) Rchb. f. and *Chrysanthemum morifolium* (Ramat.) Hemsl. The multifunctional therapeutic application of arjunolic acid has already been documented in terms of its various biological functions, including anti-oxidant, anticholinesterase, anti-tumor, and anti-asthmatic effects [122]. Arjunolic acid could ameliorate lipid accumulation, inhibit steatosis, and reduce blood fat. The upregulation of PPARγ expression may be an important molecular mechanism of action [123].

#### 3.4.12. Ganoderic Acid A

Ganoderic acid A is a triterpenoid compound, extracted from *Ganoderma lucidum* (Curtis) P. Karst. Ganoderic acid A shows a variety of pharmacological activities, such as reducing blood lipid levels, lowering blood pressure, protecting the liver, and regulating liver function [124]. Ganoderic acid A could improve NAFLD by regulating the levels of signaling events involved in free fatty acid production, lipid oxidation, and liver inflammation. Previous research confirms that the alteration of the expression of signaling events in the AMPK-mediated pathway is its key mechanism [125].

#### 3.4.13. Ilexgenin A

Ilexgenin A, a pentacyclic triterpenoid, is among the main bioactive compounds in *Quercus aliena* Blume and Ilex brachyphylla (Hand.-Mazz.) S. Y. Hu. Ilexgenin A showed beneficial effects on rats with NAFLD by lowering their liver weight, regulating lipid metabolism, and ameliorating steatosis in the liver. In addition, Ilexgenin A synergistic effect on other drugs in NAFLD has been further explored. Ilexgenin A enhances the effects of simvastatin and combinations of drugs on NAFLD without changes in the pharmacokinetics of simvastin [126].

#### 3.4.14. Rotundic Acid

Rotundic acid is a natural pentacyclic triterpene compound in *Ilex rotunda* Thunb. with many pharmacological activities. As a single pure compound, rotundic acid’s therapeutic effect and mechanism have been reported in relation to liver diseases such as hepatocellular carcinoma [127]. The value of rotundic acid in NASH treatment is gradually being recognized. The sterol regulatory element binding proteins-1c (SREBP-1c) and sterol-coA desaturase-1 (SCD-1) play an important role in the process of fatty acid anabolism. Rotundic acid could attenuate the symptoms of NASH, and the mechanism of action was through downregulating the expression of the SREBP-1c/SCD1 signaling pathway [128].

#### 3.4.15. Saikosaponin D

Saikosaponin D, the epoxy ether-type pentacyclic triterpenoid compound extracted from Radix bupleuri, exerts various pharmacological properties, with its anti-inflammatory activity being among the key benefits [129]. Saikosaponin D has a hepatoprotective effect in NAFLD via its anti-inflammatory action and ability to act as an anti-oxidant [130]. However, the specific mechanism of saikosaponin D is still unclear and needs to be further studied.

### 3.5. Tetraterpenoids

Tetraterpenoids are composed of eight isoprene units and C40 polyene. All carotenoids are derivatives of tetraterpenes, and are thus produced from eight isoprene molecules (four terpene units) [131]. In total, our study identified six tetraterpenoids capable of treating NAFLD.

#### 3.5.1. Lycopene

Lycopene belongs to the tetraterpene carotenoid family and is found in red fruits and vegetables, such as *Solanum lycopersicum* L., *Citrullus lanatus* (Thunb.) Matsum. and Nakai, *Citrus aurantium* (Lour.) Engl., and *Daucus carota* var. sativa Hoffm. Eleven conjugated double bonds predetermine the anti-oxidant properties of lycopene and its ability to scavenge lipid peroxyl radicals, reactive oxygen species, and nitric oxide [132]. Lycopene has numerous biological activities, such as anti-cancer, anti-oxidant, cardioprotective, anti-inflammatory, anti-platelet aggregative, and anti-hypertensive action [133]. Carotenoid metabolism in animals is mainly dependent on two enzymes—BCMO1 and β-carotene-9′,10′-dioxygenase (BCO2)—and lycopene is the same. Lycopene feeding has widespread effects on hepatic metabolism, stress, nuclear receptors, and nuclear coregulator gene expression, of which some are substantially dependent upon the BCO2 genotype. A total of 19 genes were affected by lycopene feeding [134]. NAFLD is strongly associated with mesenteric adipose tissue (MAT), and represents an immune dialogue between the gut and liver. The study of the enterohepatic immune axis provides evidence that MAT inflammation contributes to the pathology of NAFLD. The lycopene-mediated ability to prevent hepatic steatosis was associated with modulations in MAT lipid metabolism. Lycopene reduced steatosis by increasing MAT fatty acid utilization, and upregulating PPARa-inducible genes may be among its functions [135].

Lycopene prevents the progression of lipotoxicity-induced non-alcoholic steatohepatitis by decreasing oxidative stress in mice. The intrinsic mechanism of action is lycopene decreasing LPS-/IFN-γ-/TNFα-induced M1 marker mRNA levels in peritoneal macrophages, as well as the TGF-β1-induced expression of fibrogenic genes in a stellate cell line [136]. The hepatoprotective and anti-oxidant effects of lycopene on NAFLD, downregulating the expression of TNF-α and CYP2E1, may be among the mechanisms of action [137]. Lycopene reduced hepatic steatosis in mice fed a high-fat diet by upregulating miR-21, identifying FABP7 as a novel target of miR-21 [138]. In addition, lycopene can reduce multiple risk factors of non-alcoholic fatty liver disease. Studies have confirmed that lycopene can reduce the damage to the liver caused by obesity [139], smoking [140], hypertension [141], diabetes [142], hyperlipidemia [143], and other risk factors, which is conducive to curbing the occurrence and development of NAFLD. Of course, many diseases could benefit from the effects described above.

The synergistic effect of lycopene in combination with other drugs is a new development for lycopene. Luteolin and lycopene in combination can effectively ameliorate NAFLD in a “two-hit” manner through the activation of the Sirt1/AMPK pathway. The luteolin (20 μM) + lycopene (10 μM) therapeutic combination was found to be the best, and significantly improved cell viability and lipid accumulation in PA-induced HepG2 cells and primary hepatocytes [144].

#### 3.5.2. Astaxanthin

Astaxanthin is a kind of ketone carotenoid belonging to the tetraterpenoids [145]. Astaxanthin is widely found in nature (especially in shrimp, fish, crab, algae, etc.), and shows a wide range of pharmacological activities, include anti-oxidant, anti-inflammatory, anti-lipid peroxidation, and immune enhancement effects. Studies have shown that astaxanthin has important preventive and therapeutic effects on liver fibrosis, non-alcoholic fatty liver, liver cancer, and drug- and ischemia-induced liver injury, and has potential as a therapeutic agent in both healthy and diseased livers [146]. One animal study found that the expression of 8848 genes was associated with NASH in mice. Among these genes, 1137 were significantly up- or downregulated by astaxanthin [147]. Previous research suggests that astaxanthin might be a novel and promising treatment for NAFLD, and can effectively prevent and treat the disease at multiple stages. The increased hepatic expression of endogenous anti-oxidant genes is an effective way for astaxanthin to intervene in NAFLD. In an astaxanthin-supplemented liver, NRF-2 mRNA expression doubled, and the expression of its target endogenous anti-oxidant genes increased [148]. Fibrosis results from the dysregulation of fibrogenesis in hepatic stellate cells (HSCs). Astaxanthin prevents TGFβ1-induced pro-fibrogenic gene expression by inhibiting Smad3 activation in hepatic stellate cells [149]. Vitamin E has become a standard treatment for NASH. Astaxanthin displays multiple functions in the inhibition of NASH progression via modulating intrahepatic immunity comparison with vitamin E [150]. Liver macrophages play a central role in inflammatory cell infiltration and immune response in NASH. The beneficial effects of astaxanthin were attributable in part to both the decreased hepatic recruitment of T cells and macrophages, as well as an M2-dominant polarization of macrophages/Kupffer cells to attenuate hepatic inflammation and fibrosis [151]. Mitochondria regulate hepatic lipid metabolism, cell death, and oxidative stress. Astaxanthin attenuated hepatocyte damage and mitochondrial dysfunction in NAFLD by upregulating the FGF21/PGC-1α pathway [152].

#### 3.5.3. β-Cryptoxanthin

β-Cryptoxanthin, a xanthophyll carotenoid, has been isolated from a variety of sources, including orange, papaya, egg yolk, butter, and apples. β-cryptoxanthin has been shown to contribute to the improvement of NAFLD through a multifaceted approach, including improved insulin resistance, the suppression of inflammation and oxidative stress, a reduction in macrophages and a shift of their subsets, and the control of lipid metabolism by PPAR family activation [153]. A study recruited 92 NAFLD outpatients for a 12-week, single-center, parallel-group, double-blind RCT, and confirmed that a hypocaloric high-protein diet supplemented with β-cryptoxanthin safely and efficaciously improves NAFLD [154]. This clinical study also revealed that an energy-restricted HPD supplemented with BCX more efficaciously alleviates oxidative stress and inflammation in NAFLD as compared with a standard energy-restricted diet [155]. β-Carotene oxygenases 1 and 2 (BCO1 and BCO2) are the enzymes that metabolize carotenoids. β-cryptoxanthin feeding mitigates high-refined-carbohydrate diet (HRCD)-induced NAFLD in both wild-type (WT) and BCO1^−/−^/BCO2^−/−^ double-knockout (DKO) mice through different mechanisms in the liver–mesenteric adipose tissues axis, depending on the presence or absence of BCO1/BCO2 [156]. The consumption of β-cryptoxanthin possibly prevents NASH, which contributes to the anti-oxidative stress, anti-inflammation, immunoregulatory, and fibrosis-suppressing effects of β-cryptoxanthin [157]. The expression of lipopolysaccharide-inducible and/or TNF-α-inducible genes was suppressed by cryptoxanthin, probably via the inhibition of macrophage activation. NAFLD can be defined as lipotoxic liver injury and progression to NASH. β-cryptoxanthin prevents and reverses insulin resistance and steatohepatitis, at least in part, through an M2-dominant shift in macrophages/Kupffer cells in a lipotoxic model of NASH [158].

#### 3.5.4. β-Carotene

β-carotene is a common carotenoid that is important for human health. Major sources of β-carotene in the human diet are primarily green leafy vegetables, carrots, apricots, sweet potatoes, red palm oil, mature squashes, pumpkins, and mangoes [159]. β-carotene might be a promising preventive and protective nutrient for fatty liver disease, and has been reported to alleviate hepatic steatosis (SS), inflammation, and fibrosis in in vivo and in vitro studies [160]. In the dietary carotenoids and NAFLD among US Adults across 2003–2014 study, it was confirmed that higher intake and serum levels of most carotenoids were associated with lower odds of having NAFLD [161]. A community-based cross-sectional study involving total of 2935 participants aged 40–75 years found that higher levels of α-carotene, β-carotene, lutein + zeaxanthin, and total carotenoids were significantly associated with a decrease in the degree of NAFLD [162]. β-carotene is beneficial for the treatment and prevention of NAFLD, and can enhance the therapeutic effect of other drugs on NAFLD. The combination of rosuvastatin with β-carotene is more effective than rosuvastatin alone [163]. α-carotene was recognized as having therapeutic value for NAFLD, but α-carotene has consistently been included in broader carotenoid studies, with a lack of more specific work.

#### 3.5.5. Lutein

Lutein and its isomers, zeaxanthin and meso-zeaxanthin, are xanthophyll carotenoids found commonly in green leafy vegetables, avocados, and eggs, which play significant roles in human health [164]. Lutein is involved in the treatment of NAFLD by regulating the expression of the key factors related to insulin signaling and lipid metabolism in the liver. Lutein supplementation could ameliorate hepatic lipid accumulation and insulin resistance induced by a HFD, possibly via the activation of the expression of SIRT1/PPAR-α and other key factors in insulin signaling pathways [165].

#### 3.5.6. Cycloastragenol

Cycloastragenol, a tetracyclic triterpenoid compound, is a secondary metabolite isolated from *Astragalus membranaceus var. mongholicus* (Bunge) P. K. Hsiao, and has a wide spectrum of pharmacological functions, which are attracting attention in the research community. It has been shown to have anti-aging, anti-oxidation, anti-inflammation, anti-cancer, and cardiovascular-protective effects [166]. Nuclear receptor FXR, although known as a bile acid receptor, is also related to liver lipid metabolism and glucose metabolism. Cycloastragenol alleviates hepatic steatosis, reduces lipid accumulation, and lowers blood glucose in NAFLD via the stimulation and enhancement of the FXR signaling pathway [167].

**Table 1 molecules-28-00272-t001:** Introduction to the basic information of terpenoids.

Subclass	Compound	Molecular Formula	Weight (g/mol)	Sources
Monoterpenoid	Paeoniflorin	C_23_H_28_O_11_	480.5	*Paeonia albiflora* Pallas
Iridoid	Catalpol	C_15_H_22_O_10_	362.3	*Rehmannia glutinosa* (Gaertn.) Libosch. ex Fisch. and C. A. Mey.
Iridoid	Geniposide	C_17_H_24_O_10_	388.4	*Gardenia jasminoides* J. Ellis
Iridoid	Genipin	C_11_H_14_O_5_	226.2	*Gardenia jasminoides* J. Ellis
Iridoid	Sweroside	C_16_H_22_O_9_	358.3	*Swertia bimaculata* (Siebold and Zucc.) Hook. f. and Thomson ex C. B. Clarke
Iridoid	Swertiamarin	C_16_H_22_O_10_	374.3	*Swertia bimaculata* (Siebold and Zucc.) Hook. f. and Thomson ex C. B. Clarke
Iridoid	Aucubin	C_15_H_22_O_9_	346.3	*Eucommia ulmoides* Oliv., *Plantago asiatica* L., *Scrophularia ningpoensis* Hemsl.
Iridoid	Gentiopicroside	C_16_H_20_O_9_	356.3	*Gentiana scabra* Bunge
Monoterpenoid	Geraniol	C_10_H_18_O	154.3	*Elsholtzia ciliata (Thunb.) Hyl.*, *Murraya exotica L.*, *Rosa rugosa Thunb.*
Sesquiterpenoid	Curcumol	C_15_H_24_O_2_	236.4	*Curcuma phaeocaulis* Valeton
Sesquiterpenoid	β-patchoulene	C_15_H_24_	204.4	*Pogostemon cablin* (Blanco) Benth.
Sesquiterpenoid	β-caryophyllene	C_15_H_24_	204.4	*Rosmarinus officinalis* L., *Cinnamomum cassia* (L.) D. Don, *Ocimum basilicum* L., *Lavandula angustifolia* Mill.
Diterpenoid	Carnosic acid	C_20_H_28_O_4_	332.4	*Rosmarinus officinalis* Linnaeus, *Salvia japonica* Thunb.
Diterpenoid	Ginkgolide A	C_20_H_24_O_9_	408.4	*Ginkgo biloba* L.
Diterpenoid	Ginkgolide B	C_20_H_24_O_10_	424.4	*Ginkgo biloba* L.
Diterpenoid	Acanthoic acid	C_20_H_30_O_2_	302.5	*Eleutherococcus senticosus* (Rupr. and Maxim.) Maxim.
Diterpenoid	Dehydroabietic acid	C_20_H_28_O_2_	300.4	*Pinus massoniana* Lamb., *Picea asperata* Mast.
Triterpenoid	Ginsenoside Rg1	C_42_H_72_O_14_	801	*Panax ginseng* C. A. Mey.
Triterpenoid	Ginsenoside Rg2	C_42_H_72_O_13_	785	*Panax ginseng* C. A. Mey.
Triterpenoid	Ginsenoside Rh1	C_36_H_62_O_9_	638.9	*Panax ginseng* C. A. Mey.
Triterpenoid	Ginsenoside Rb1	C_54_H_92_O_23_	1109.3	*Panax ginseng* C. A. Mey.
Triterpenoid	Ginsenoside Rb2	C_53_H_90_O_22_	1079.3	*Panax ginseng* C. A. Mey.
Triterpenoid	Ginsenoside Mc1	C_47_H_80_O_17_	917.1	*Panax ginseng* C. A. Mey.
Triterpenoid	Ursolic acid	C_30_H_48_O_3_	456.7	*Gentiana scabra* Bunge, *Pseudocydonia sinensis* (Thouin) C. K. Schneid., *Eriobotrya japonica* (Thunb.) Lindl., *Pyrrosia lingua* (Thunb.) Farw. and *Plantago asiatica* L.
Triterpenoid	Betulinic acid	C_30_H_48_O_3_	456.7	*Betula platyphylla* Sukaczev
Triterpenoid	Glycyrrhizic acid	C_42_H_62_O_16_	822.9	*Glycyrrhiza uralensis* Fisch.
Triterpenoid	Glycyrrhetinic acid	C_30_H_46_O_4_	470.7	*Glycyrrhiza uralensis* Fisch.
Triterpenoid	Oleanolic acid	C_30_H_48_O_3_	456.7	*Ophiopogon japonicus* (L. f.) Ker Gawl., *Swertia leducii* Franch., *Ligustrum lucidum* W. T. Aiton
Triterpenoid	Astragaloside IV	C_41_H_68_O_14_	785	*Astragalus membranaceus var. mongholicus* (Bunge) P. K. Hsiao
Triterpenoid	Mogroside V	C_60_H_102_O_29_	1287.4	*Siraitia grosvenorii* (Swingle) C. Jeffrey ex A. M. Lu and Zhi Y. Zhang
Triterpenoid	Asiatic acid	C_30_H_48_O_5_	488.7	*Centella asiatica* (L.) Urb.
Triterpenoid	Corosolic acid	C_30_H_48_O_4_	472.7	*Eriobotrya japonica* (Thunb.) Lindl., *Actinidia chinensis* Planch., *Hippophae rhamnoides* L. *and Lagerstroemia speciosa* (L.) Pers
Triterpenoid	Arjunolic acid	C_30_H_48_O_5_	488.7	*Cymbidium goeringii* (Rchb. f.) Rchb. f., Chrysanthemum morifolium (Ramat.) Hemsl.
Triterpenoid	Ganoderic acid A	C_30_H_44_O_7_	516.7	*Ganoderma lucidum* (Curtis) P. Karst.
Triterpenoid	Ilexgenin A	C_30_H_46_O_6_	502.7	*Quercus aliena* Blume, *Ilex brachyphylla* (Hand.-Mazz.) S. Y. Hu
Triterpenoid	Rotundic acid	C_30_H_48_O_5_	488.7	*Ilex rotunda* Thunb.
Triterpenoid	Saikosaponin D	C_42_H_68_O_13_	781	*Radix bupleuri*
Tetraterpenoid	Lycopene	C_40_H_56_	536.9	*Solanum lycopersicum* L., *Citrullus lanatus* (Thunb.) Matsum. and Nakai, *Citrus aurantium* (Lour.) Engl., *Daucus carota var*. sativa Hoffm.
Tetraterpenoid	Astaxanthin	C_40_H_52_O_4_	596.8	Shrimp, fish, crab, algae, etc.
Tetraterpenoid	β-cryptoxanthin	C₄₀H₅₆O	552.9	Orange, papaya, egg yolk, butter, apples, etc.
Tetraterpenoid	β-carotene	C_40_H_56_	536.9	Carrots, apricots, sweet potatoes, mature squashes, pumpkins, mangoes, etc.
Tetraterpenoid	Lutein	C_40_H_56_O_2_	568.9	Spinach, kale, yellow carrots, etc.
Tetraterpenoid	Cycloastragenol	C_30_H_50_O_5_	490.7	*Astragalus membranaceus var. mongholicus* (Bunge) P. K. Hsiao

**Table 2 molecules-28-00272-t002:** The effects and mechanisms of 43 terpenoids on NAFLD, from recent studies.

Compound	Animal/Cell Model	Dosage (mg/kg/d; μM/24 h)	Target/Pathways/Mechanism	Effects	Reference
Paeoniflorin	HFD-induced NAFLD mice	0.05% in diet	Activation of the CD36/AMPK signaling pathway	Reduced body weight, improved insulin resistance, anti-inflammatory, inhibition of lipid accumulation, attenuated hepatic adipose infiltration	[24]
	HFD-induced NAFLD rats	20	Regulation of the IRS/Akt/GSK3β signaling pathway	Inhibition of lipid accumulation, improved insulin resistance, anti-oxidative stress, liver protection	[25]
	Fructose-induced metabolic syndrome rats	10, 20, 40	Activation of the AMPK signaling pathway	Inhibition of hepatic lipid accumulation, improved insulin resistance, inhibition of hepatic steatosis, inhibition of hepatic lipogenesis, promotion of fatty acid oxidation	[26]
	HFD-induced NAFLD rats	20, 60, 100	Inhibition of the ROCK/NF-κB signaling pathway	Anti-inflammatory, ameliorated hepatic steatosis, reduced lipids	[27]
Catalpol	HFD-induced NAFLD mice; PA-induced HepG2 cells	100, 200, 400; 100, 200, 400	Activation of the AMPK signaling pathway	Ameliorated hepatic steatosis, reduced body weight, inhibition of lipid accumulation	[29]
	HFD-induced NAFLD mice; PA-induced HepG2 cells	100; 10 μg/mL/24h	/	Inhibition of autophagy, ameliorated hepatic steatosis, reduced liver weight, reduced liver fat	[31]
	(LDLr^−/−^)+ HFD-induced NAFLD mice	100	Regulation of the p66shc/cytochrome C signaling pathway	Attenuated liver injury, anti-oxidative stress, inhibition of hepatic steatosis, anti-apoptosis	[32]
	HFD-induced NAFLD mice	100, 200, 400	/	Ameliorated hepatic steatosis, reduced body weight, improvement of lipid metabolism disorders, inhibition of lipid accumulation, anti-inflammatory, anti-apoptosis	[33]
Geniposide	(PA + OA)-induced HepG2 cells	0, 65, 130, 260, 390, 520 μmol/L/24 h	Upregulation of the Nrf2/AMPK/mTOR signaling pathways	Inhibition of lipid accumulation, anti-oxidative stress, anti-inflammatory	[35]
	HFD-induced NAFLD rats	25, 50, 100	Increased expression of PPARa gene	Ameliorated hepatic steatosis, anti-oxidative stress	[36]
Genipin	HFD-induced NAFLD mice; (PA + OA)-induced cells primary hepatocytes of mice	5, 20; 5, 20	Regulation of the miR-142a-5p/SREBP-1c axis	Reduced body weight gain, increased locomotor activity, improved insulin resistance, alleviated hyperlipidemia, inhibition of lipid accumulation	[38]
	HFD-induced NAFLD mice	5, 20	Suppressed UCP2	Reversed liver damage, anti-pyroptosis	[39]
Sweroside	HFD-induced NAFLD mice	60, 120, 240	Increases expression of PPARa gene	Reduced body weight, improved insulin resistance, inhibited hepatic steatosis, anti-inflammatory	[42]
	MCD diet-induced NAFLD mice	5, 30	Suppressed activation of the NLRP3 inflammasome	Anti-inflammatory, inhibition of hepatic lipid accumulation, anti-fibrosis	[43]
Swertiamarin	HFD-induced NAFLD mice; LPSO-induced murine monocytic cells	10, 100; 1, 10, 50	Suppressed activation of the p38 MAPK and NF-κB signaling pathways	Ameliorated hepatic steatosis, anti-inflammatory, reduced body weight, improved insulin resistance	[45]
Aucubin	Tyloxapol-induced NAFLD mice	10, 20, 40	Activation of the Nrf2/HO-1 and AMPK signaling pathways	Inhibition of lipid accumulation, anti-oxidative stress, anti-inflammatory	[47]
Gentiopicroside	Tyloxapol-induced NAFLD mice; (PA + OA)-induced HepG2 cells	20, 40, 80; 0, 4, 20, 100, 200, 500	Upregulation of the Nrf2 signaling pathway	Inhibition of hepatic lipid accumulation, anti-oxidative stress	[49]
Geraniol	MCD-induced NAFLD rats	25, 100, 200	/	Inhibition of hepatic lipid accumulation, anti-fibrosis, anti-apoptosis, anti-inflammatory, anti-oxidative stress	[50]
Curcumol	HFD-induced NAFLD mice	15, 30, 60	Regulation of the YAP/NCOA4 signaling pathway	Inhibition of hepatocyte senescence, suppressed ferritinophagy	[52]
	HFD-induced NAFLD rats	25, 50, 100	Regulation of the TLR4, TAK1, and NF-κB/P65 signaling pathways	Anti-inflammatory, improved liver function, anti-fibrosis, anti-apoptosis	[53]
β-patchoulene	HFD-induced NAFLD rats; (FFA + PA + OA)-induced HepG2 cells	10, 20, 40; 40	Activation of the AMPK signaling pathway	Inhibition of hepatic lipid accumulation, improved insulin resistance, ameliorated hepatic steatosis, inhibition of hepatic lipogenesis, promotion of fatty acid oxidation	[54]
	HFD-induced NAFLD rats	10, 20, 40	Activation of the CD36/AMPK signaling pathway	Reduced body weight, reversed liver damage, ameliorated hepatic steatosis, anti-oxidative stress, anti-inflammatory	[55]
β-caryophyllene	PA-induced HepG2 cells	40	Activation o the f AMPK signaling pathway	Inhibition of hepatic lipid accumulation	[56]
Carnosic acid	Obese leptin-deficient (ob/ob) mice	0.05% in diet	/	Reduced body weight, inhibition of lipid accumulation, recovered glucose tolerance	[60]
	Obese leptin-deficient (ob/ob) mice	0.01, 0.02% in diet	/	Improved glucose tolerance, inhibition of lipid accumulation, reduced body weight	[61]
	HFD-induced NAFLD mice	15	Upregulation of MARCKS expression/impairment of the PI3K/AKT and NLRP3 inflammasome signaling pathways	Inhibition of lipid accumulation, anti-inflammatory, improved insulin resistance	[63]
	HFD-induced NAFLD mice, PA-induced human L0246 hepatic cell	30, 60; 10	Inhibition of the miR-34a/SIRT1/p66shc signaling pathway	Inhibition of lipid accumulation, anti-apoptosis	[64]
Ginkgolide A	HFD-induced NAFLD mice; NEFA-induced HepG2 cell	5; 0, 10, 50, 100		Inhibition of lipid accumulation, induced cellular lipoapoptosis, anti-inflammatory, reduced body weight	[69]
Ginkgolide B	HFD-induced NAFLD mice	0.1 % in diet	Activation of pregnane X receptor	Reduced body weight, ameliorated hepatic steatosis	[70]
	HFD-induced NAFLD mice; (PA + OA)-induced HepG2 cells	20, 30; 4, 8, 16	Increased Nrf2 expression	Anti-oxidative stress, reduced body weight, inhibition of lipid accumulation, anti-inflammatory	[71]
Acanthoic acid	Modified Lieber–DeCarli diet-induced mice	20, 40	Via FXR–LXR axis	Inhibition of hepatic lipid accumulation, anti-fibrosis, regulation of fatty acid synthesis	[75]
Dehydroabietic acid	HFD-induced NAFLD mice; OA-induced HL7702 cells	10, 20; 2.5, 5, 10	Activation of the Keap1/Nrf2-ARE signaling pathway	Reduced blood lipid, inhibition of ferroptosis	[77]
Ginsenoside Rg1	(PA + OA)-induced HepG2 cells	25, 50	Regulation of PPAR ɑ and PPAR γ expression	Inhibition of lipid accumulation, ameliorated hepatic steatosis	[80]
	D-galactose-induced fatty liver disease mice	40	Upregulation of FOXO1 gene	Anti-inflammatory, inhibition of lipid accumulation	[81]
	HFD-induced NAFLD mice	20, 40	/	Anti-inflammatory, reduced body weight, alleviated ER stress	[82]
	HSD-induced NAFLD rats	100	Regulation of Atf3 and Acox2 gene	Reduced body weight, reduced blood lipid, alleviated hepatic steatosis	[83]
Ginsenoside Rg2	HFD-induced NAFLD mice	2.5, 5, 10	Regulation of the SIRT1 signaling pathways	Improvement of lipid and glucose disorders, anti-oxidative stress, anti-apoptosis, inhibition of lipid accumulation	[85]
Ginsenoside Rh1	HFD-induced NAFLD rats	3	/	Anti-fibrotic	[86]
Ginsenoside Rb1	HFD-induced NAFLD mice	10	Activation of PPAR-γ expression	Reduced body weight, improved glucose metabolism, inhibition of lipid accumulation, anti-apoptosis	[87]
Ginsenoside Rb2	db/db mice, OA-induced HepG2 cells	10; pretreated with 0.1, 1, 10, 50, 100 μmol/L/4h	Regulation of the SIRT1 and AMPK signaling pathways	Alleviated hepatic steatosis, improved glucose tolerance, regulation of hepatic autophagy, inhibition of lipid accumulation	[88]
Ginsenoside Mc1	HFD-induced NAFLD mice	10	/	Alleviated ER stress, anti-apoptosis, improved insulin resistance, alleviated hepatic steatosis	[89]
Ursolic acid	HFD-induced NAFLD rats; human normally hepatic immortal cell line HL-7702	0.125, 0.25, 0.5% in diet; 0, 25, 50, 100	Regulation of PPAR ɑ expression	Reduced body weight, alleviated hepatic steatosis, improved metabolic disorders, improved insulin resistance, anti-inflammatory, anti-oxidative stress	[91]
	db/db mice (type 2 diabetic mouse model); PA-induced HepG2 cells	0.14% in diets; 10–30	/	Inhibition of lipid accumulation, alleviated ER stress, reduced liver weight and reduced liver injury, alleviated hepatic steatosis	[92]
	DMSO-induced human hepatocellular carcinoma cell	5, 10	Regulation of LXRα	Inhibition of lipid accumulation, alleviated hepatic steatosis, reduced blood lipids	[93]
	Fructose induced NAFLD newborn rats	10	/	Inhibition of lipid accumulation	[94]
Betulinic acid	HFD-induced NAFLD rats	0.1% in diet	Regulation of the PERK/EIF2α/ATF4/CHOP signaling pathway	Enhanced energy expenditure, modulation of bile acids, alleviated hepatic steatosis, anti-inflammatory, alleviated ER stress	[95]
	HFD-induced NAFLD rats; insulin-resistant HepG2 cells	5, 10; 10–40	Regulation of the AMPK–mTOR–SREBP signaling pathway	Inhibition of lipid accumulation	[96]
	HFD-induced NAFLD mice; (PA + OA)-induced mice primary hepatocytes	150; 10	Inhibition of the YY1/FAS signaling pathway	Inhibition of lipid accumulation, alleviated fatty acid synthesis, anti-fibrosis, anti-inflammatory, inhibition of excessive lipogenesis	[97]
Glycyrrhizic acid	MCD diet-induced NAFLD mice	12.5, 25, 50	/	Inhibition of lipid accumulation, anti-inflammatory, anti-fibrosis, improved lipid metabolism	[98]
	HFD-induced NAFLD mice	15, 30, 60	/	Inhibition of lipid accumulation, reduced hepatic lipogenesis, reduced body weight, ameliorated hepatic steatosis, reduced serum glucose, improved glucose tolerance and insulin sensitivity	[99]
	MCD diet-induced NAFLD mice	30, 50	/	Inhibition of lipid accumulation, modulation of bile acids, anti-inflammatory, anti-fibrosis	[101]
Glycyrrhetinic acid	HFD-induced NAFLD mice	15, 30, 60	/	Regulation of vitamin A metabolism, protection against hepatic injury	[103]
	HFD-induced NAFLD mice	60	Suppression of HNF4α	Reduced blood glucose, improved insulin resistance	[104]
Oleanolic acid	HFD-induced NAFLD rats	25, 50, 100	Inhibition of LXRs, activation of the AMPK pathways	Alleviated hepatic steatosis, anti-inflammatory, anti-oxidative stress, improved insulin resistance	[107]
	HFHCD-induced NAFLD rats	80	/	Decreased blood lipids, anti-oxidative stress, reversed hepatic degeneration	[109]
Astragaloside IV	(PA + OA)-induced HepG2 cells and primary murine hepatocytes	50–200	Activation of the AMPK signaling pathway	Inhibition of lipid accumulation, inhibition of lipogenesis, alleviated ER stress	[113]
	High-concentration insulin or OA-induced HepG2 cells	25.6, 51.2, 102.4	Inhibition of protein tyrosine phosphatase 1B	Improved insulin resistance, inhibition of lipid accumulation	[114]
Mogroside V	HFD-induced NAFLD mice	400, 800	Upregulation of pAMPK expression	Inhibition of lipid accumulation, anti-inflammatory, anti-oxidative stress	[116]
	HFD-induced NAFLD mice; (PA + OA)-induced human LO2 cells	25, 50, 100; 15, 30, 60,120	Activation of the AMPK signaling pathway	Inhibition of lipid accumulation, ameliorated hepatic steatosis	[117]
Asiatic acid	HFD-induced NAFLD rats	4, 8	Inhibition of the ERS signaling pathway	Inhibition of lipid accumulation, anti-inflammatory, anti-oxidative stress	[119]
Corosolic acid	HFD + CCl4-induced NAFLD mice; FFA + OA + PA-induced HepG2 cells	10, 20; 5, 10, 20	Regulation of the TGF-β1/Smad2, NF-κB, and AMPK signaling pathways	Inhibition of lipid accumulation, anti-inflammatory, anti-fibrosis	[121]
Arjunolic acid	HFD-induced NAFLD rats; (PA + OA)-induced HepG2 cells	100, 200; 12.5, 50	Upregulation of PPARγ expression	Inhibition of lipid accumulation, ameliorated hepatic steatosis, reduced blood lipids	[123]
Ganoderic acid A	HFD-induced NAFLD rats	20, 40	Activation of the AMPK signaling pathway	Inhibition of lipid accumulation, anti-inflammatory, reduced live weight	[125]
Ilexgenin A	HFD-induced NAFLD rats	80	/	Ameliorated hepatic steatosis, hypolipidemic, anti-inflammatory, enhanced effects of simvastatin	[126]
Rotundic acid	HFD-induced NAFLD rats; insulin-induced primary hepatocytes	10, 30, 100; 6.25–200	Downregulation of the SREBP-1c/SCD1 signaling pathway	Inhibition of lipid accumulation, improved dyslipidemia, protection against hepatic injury, anti-inflammatory, inhibition of excessive lipogenesis	[128]
Saikosaponin D	TAA-induced liver injury mice; HFD-induced NAFLD mice	2	/	Reduced blood lipids, anti-oxidative stress, anti-inflammatory	[130]
Lycopene	HFD-induced NAFLD mice	100, 1000	Upregulation of PPARa-inducible genes	Ameliorated hepatic steatosis	[135]
	HFD-induced NAFLD mice	0.004, 0.012% in diet	/	Inhibition of lipid accumulation, improved insulin resistance, anti-fibrosis, anti-inflammatory, anti-oxidative stress	[136]
	HFD-induced NAFLD rats	5, 10, 20	Downregulated expression of TNF-ɑ and CYP2E1	Improved lipid profiles, reduced lipid peroxides, reduced blood lipids	[137]
	HFD-induced NAFLD mice	0.05% in diet	microRNA-21-induced downregulation of fatty-acid-binding protein 7	Ameliorated hepatic steatosis, inhibition of hepatic lipid accumulation	[138]
	HFD-induced NAFLD rats	20	/	Ameliorated hepatic steatosis, reduced liver weight, reduced blood lipids	[140]
Astaxanthin	HFD-induced NAFLD mice	0.02% in diet	Inhibition o the f eIF-2 signaling pathway	Inhibition of lipid accumulation, anti-inflammatory, anti-fibrosis, anti-oxidative stress	[147]
	HFD-induced NAFLD mice	0.003, 0.01, 0.03% in diet	/	Alleviated hepatic steatosis, anti-inflammatory, anti-oxidative stress	[148]
	Hepatic stellate cells from humans and mice	0–200	Inhibition of the TGFβ1–Smad3 signaling pathway	Anti-oxidative stress, anti-fibrosis	[149]
	HFD-induced NAFLD mice	80	/	Alleviated hepatic steatosis, anti-inflammatory, anti-oxidative stress	[150]
	HFD-induced NAFLD mice	0.0067, 0.02% in diet	/	Inhibition of lipid accumulation, alleviated hepatic steatosis, improved glucose intolerance, improved insulin resistance, anti-inflammatory, anti-fibrosis	[151]
	HFD-induced NAFLD mice; human liver cell line	10, 30, 60; 30, 60, 90	Upregulating the FGF21/PGC-1α signaling pathway	Inhibition of lipid accumulation, anti-oxidative stress, anti-apoptosis, anti-inflammatory, anti-fibrosis, attenuated mitochondrial dysfunction	[152]
β-cryptoxanthin	HRCD + DKO-induced NAFLD mice	10	Activation of the SIRT1/AMPK signaling pathway	Inhibition of lipid accumulation, alleviated hepatic steatosis, increased cholesterol efflux	[156]
	HFD-induced NASH mice	0.003% in diet	/	Anti-inflammatory, anti-oxidative stress, anti-fibrosis, alleviated hepatic steatosis, inhibition of lipid accumulation	[157]
	HFD-induced NASH mice	0.003% in diet	/	Anti-inflammatory, anti-oxidative stress, anti-fibrosis, alleviated hepatic steatosis, inhibition of lipid accumulation, improved liver dysfunction	[158]
β-carotene	HFD-induced NAFLD rats	70	/	Alleviated hepatic steatosis, anti-inflammatory	[163]
Lutein	HFD-induced NAFLD rats	0, 12.5, 25, 50	Activation of the SIRT1/PPAR-α signaling pathway	Reduced body weight, alleviated hepatic steatosis, improved insulin resistance	[165]
Cycloastragenol	HFD-induced NAFLD mice; FXR deletion HepG2 cells	0.1% in diet; 25	Regulation of the FXR signaling pathway	Alleviated hepatic steatosis, inhibition of lipid accumulation, reduced blood glucose, anti-oxidative stress	[167]

## 4. Conclusions

The data reported in this review highlight that, despite the very large number of studies investigating the health effects of terpenoids in humans, triterpenoids have been the main focus, with 20 triterpenoids proving effective in the treatment of NAFLD. Different classes of terpenoids have been shown to be present in different amounts in NAFLD treatment, and can be ranked as follows: triterpenoids > monoterpenoids > tetraterpenoids > diterpenoids > sesquiterpenoids. This is because compared with other terpenoids, triterpenoids are more diverse, widely distributed, and abundant. In particular in the 21st century, the diversity and importance of the biological activities of triterpenes have attracted much attention and have become a popular area of natural medicinal chemistry research. In addition, their effects may be related to the unique side chains of these triterpenes, which may facilitate the formation of links with the target proteins and thus exhibit better activity. It is necessary to conduct some research on molecular docking technology to further evaluate the affinity of ligand binding to the target protein receptor, so as to better explain the systematic structure–activity relationship.

The pathogenesis of NAFLD is complex and implicates a cross talk between various metabolically active sites. Experimental models play a crucial role in elucidating the pathophysiology of diseases and the pharmacological effects of the drug. In this review, animal models and cell models are the main methods of study. In order to study the pathogenesis of non-alcoholic fatty liver disease and the effect of drug treatment, transgenic animals, chemical-induced animals, and three high-fat-diet-induced experimental animal models are often used as research models for NAFLD. The animal model of fatty liver induced by a high-fat diet has similar pathological characteristics to those in humans, with a high success rate and low mortality. The method is simple, easy to repeat, and has become the main choice for drug research. Mice and rats have been used most frequently in NAFLD modeling and therefore constitute the main focus of animal model research. Although it is easy to establish NAFLD models in rats and mice, long-term drug observation cannot be achieved, which has also led to the lack of studies on the development of NALFD to end-stage cirrhosis. The cell lines used to establish cell models include primary hepatocytes from animals and humans, immortalized cell lines, and hepatoid cells derived from induced pluripotent stem cells. HepG2 is a popular hepatic cell line, used in a wide range of studies. As a class of nontumorigenic cells, with high proliferation rates and an epithelial-like morphology, it performs many differentiated hepatic functions. The HepG2 cell manufacturing model induced by palmitate and oleic acid is a common choice of NAFLD cell model. The induced pluripotent hepatocyte technology still has some problems, such as the low success rate of inducing cell transformation and easy genetic mutation in the process of induction, which limit its wide application. This review found that animal and cell experiments are often used to study the effects of terpenoids on NAFLD, and their complementarity can better reveal the mechanism of action and efficacy characteristics of terpenoids.

The different doses and methods of administration in these studies are an issue worth exploring. In animal studies, there are generally two ways to administer a drug: one is to administer a certain amount of the drug, and the other is to add it to the food in a certain percentage. Of the 76 animal modeling studies, 15 were conducted with a diet supplemented with terpenoids. Most of them use the method of administration according to a certain dose, which is more convenient for calculation and statistics, and can be easily controlled. The dosage of terpenoids in these studies was different, with even the dosage of one terpene compound being different in different experiments. In fact, different doses of terpenoids play a different role in the effective treatment of NAFLD, but there is only limited information on the effects of these different doses of terpenoids on hepatic cells. Drug administration in cell experiments mainly takes two forms: one is the direct addition into the medium, and the other is the pretreatment of the drug; the latter is less used. The cells were usually treated with different doses of terpenoids for 24 h to discover their effects on the cells. Unnecessarily high doses can increase the risk of complications and adverse effects associated with the drug. The current research is inadequate, so more studies are required to define the best dose.

Based on the above reports, there is a high probability that multiple edible terpenoids are potential candidates for managing NAFLD. Although the efficacy of terpenoids in treating NAFLD has been demonstrated in both animal and cell studies, the efficacy of most terpenoids has not been supported by clinical evidence. Some drugs have already been put to use in clinical practice, but the small number of cases in which they have been applied to date precludes an evaluation of their long-term durability. In China, oleanolic acid has been used as an over-the-counter (OTC) hepatoprotective drug for decades [168]. As early as 1993, it was used in clinical controlled trials as a classic anti-hepatitis drug [169], but there are not yet any clinical studies on NAFLD, although it is a drug with a long clinical history. Dietary carotenoids are thought to provide health benefits in terms of decreasing the risk of NAFLD and have been confirmed both in vitro and in vivo. Lycopene, astaxanthin, β-cryptoxanthin, β-carotene, and lutein are popular research topics in the field of carotenoid therapy for NAFLD. Clinical studies on these compounds are richer than those on other terpenoids. This is because these terpenoids are more commonly found in the daily diet. Ginkgo biloba extract is a popular topic in natural compound research and has a wide clinical research base. The Ginkgo biloba special extract EGb 761^®^ is currently listed in local clinical guidelines in Germany, Switzerland, and some Asian countries, including Indonesia, the Philippines, and China [170]. Although ginkgolides have been proven to be effective for NAFLD, there is no clinical research support. Glycyrrhizic acid has been developed in Japan and China as a hepatoprotective drug in cases of chronic hepatitis, such as Stronger Neo-Minophagen C and compound clycyrrhizin tablets. Licorice and its derivatives are recognized as safe by the US Food and Drug Administration (FDA). The long medicinal history and rich clinical studies of glycyrrhizic acid provide possibilities for its development and application in NAFLD. However, glycyrrhizic acid can cause a metabolic syndrome presenting as pseudohyperaldosteronism, which limits its clinical application [171]. There exists the idea that “The best way to discover a new drug is to start with an old one”—these terpenoids seem to be able to achieve clinical breakthroughs more quickly than other terpenoids.

NAFLD is not a single disease, but encompasses a range of liver diseases, from hepatic steatosis to NASH, and can eventually lead to cirrhosis and even death. Lipid metabolism disorder, insulin resistance, oxidative injury, inflammatory reaction, apoptosis, cytosis, and autophagy interact with each other to form a complex regulatory network, resulting in a series of pathological cascades of NAFLD. Terpenoids can protect the injured liver tissue by inhibiting and regulating the above factors. NAFLD is the result of a serious systemic disorder of lipid metabolism. Regulating lipid metabolism is a common effect of all terpenoids included in this review, included the inhibition of hepatic lipid accumulation, the regulation of steatosis, the inhibition of hepatic lipogenesis, and the promotion of fatty acid oxidation. Inflammation is among the features necessary for a histologic diagnosis of NASH, involved in the whole pathological process of NAFLD. A total of 28 terpenoids have been shown to play an anti-inflammatory role in NAFLD, thereby inhibiting the progression of the disease. The anti-inflammatory effects of these terpenoids also inhibit programmed cell death, liver fibrosis, and liver tissue damage. The increased production of ROS in the liver leads to lipid peroxidation/oxidative stress, which then leads to the functional dysfunction of the liver cytoplasmic body, as well as apoptosis and deterioration. A total of 19 terpenoids play a therapeutic role in NAFLD via their anti-oxidative stress activity. Insulin resistance plays a key role in the pathological process of NAFLD, with the prevalence of NAFLD being 5-fold higher in patients with diabetes compared to those without [172]. Insulin resistance is involved in the metabolic basis of NAFLD. In this review, 13 terpenoids are reported to improve insulin resistance in the study of NAFLD. In fact, a large number of terpenoids are reported to improve insulin resistance in the study of diabetes, but they have not been mentioned in the study of NAFLD. Liver fibrosis is the most important predictor of mortality in NAFLD. Untreated NASH may lead to liver fibrosis and eventually HCC. A total of 10 terpenoids have been proved to have anti-fibrotic effects, which is of great significance for the long-term treatment of NAFLD. Pyroptosis, as a novel form of pro-inflammatory programmed cell death, has become a popular research topic in the life sciences. Pyroptosis is considered to play an important role in low-grade chronic inflammation, which is thought to be the core of the pathological mechanism of NAFLD [173]. Although the involvement of pyroptosis in NAFLD has been well established, little has been written about the role of pyroptosis through this natural compound. Only one terpenoid has been reported to cause pyroptosis in NAFLD. The maintenance of iron distribution is crucial to the metabolic response of hepatocytes. The ectopic storage of iron between hepatocytes and astrocytes is a key factor in the development of non-alcoholic fatty liver disease [174]. However, only one terpenoid has been revealed to regulate iron homeostasis and protect the liver. To accelerate the application of the latest pathological mechanisms in the field of drug research, and provide new potential targets for the further drug development of terpenoids, in the future, we need to enrich the research on the effects of terpenoids on NAFLD, encouraging new ideas for drug use and disease treatment.

We found that the signaling pathways and targets of these terpenoids were consistent to a certain extent. The evolution of AMPK and its homologs enabled excellent responsivity and control of cellular energetic homeostasis [175]. The central role of AMPK in maintaining cellular energy homeostasis has made it a promising target for drugs aimed at preventing and/or treating NAFLD. The effects of 14 terpenoids on NAFLD through the AMPK signaling pathway were reported in the literature. The peroxisome proliferator-activated receptors (PPAR-α, PPAR-β/δ, and PPAR-γ) are members of the nuclear receptor super-family and play crucial roles in lipid metabolism [41]. Eight terpenoids were confirmed to exert an intervention effect through this receptor, and PPAR-α and PPAR-γ were the key to this effect, of which PPAR-α was the most frequently mentioned. PPAR-α and PPAR-γ are the master regulators of adipogenesis and adipose tissue development and both of them have specific expression in the liver. The gene expression program induced by Nrf2 transcription factor plays a critical role in cell defense responses against a broad variety of cellular stresses, including oxidative stress, metabolism, and inflammation [176]. Five terpenoids exert anti-oxidative stress, anti-inflammatory, and liver-protective effects by regulating Nrf2. In recent years, accumulating evidence has indicated that sirtuins play important roles in regulating the metabolic processes related to fatty liver diseases [177]. SIRT 1 is the most extensively studied sirtuin and is involved in fatty liver diseases. Five terpenoids regulate lipid metabolism, oxidative stress, and inflammation in the liver via SIRT 1.

The existence of these common pathways and targets provides a reference for the further improvement of the study of terpenoids in the treatment of NAFLD and promotes the study of the mechanism of action of these terpenoids.

Some terpenoids reviewed in this paper show potent activity in the treatment of NAFLD. However, current studies are restricted to animal and cell experiments, with a lack of clinical research and systematic SAR studies. Further clinical research and SAR studies with terpenoids could provide more insight into the effectiveness of these complicated pharmacological properties, enabling terpenoids to be used safely and efficiently.

## Figures and Tables

**Figure 1 molecules-28-00272-f001:**
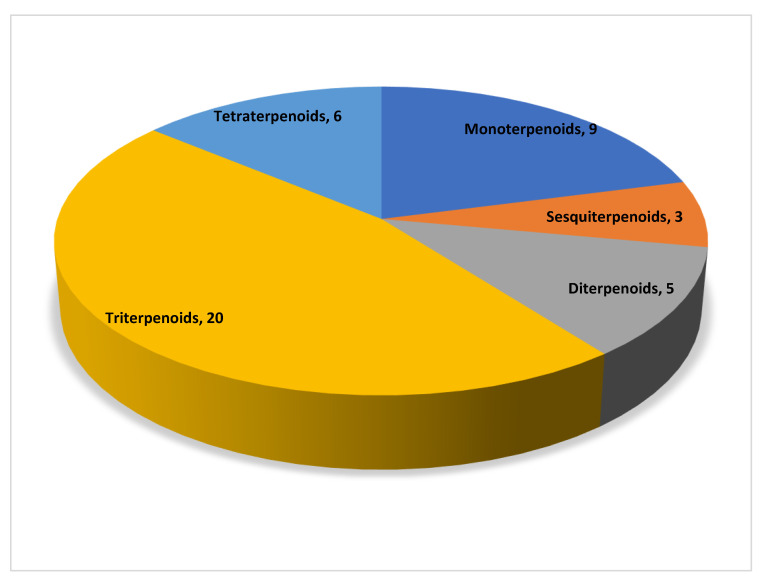
Distribution of different subclasses of terpenoids.

**Figure 2 molecules-28-00272-f002:**
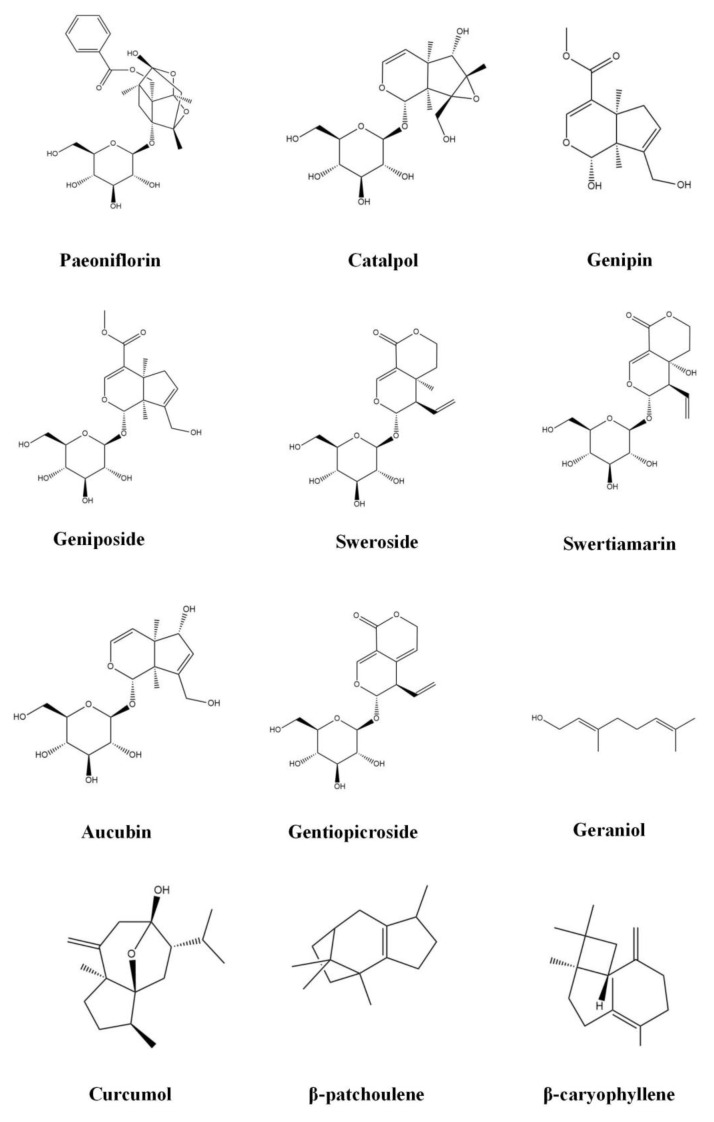
Chemical structure of terpenoids.

## Data Availability

Not applicable.

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
