# Peer review of "Terpenoids: Natural Compounds for Non-Alcoholic Fatty Liver Disease (NAFLD) Therapy"

_molecules, 2022, doi:10.3390/molecules28010272_

Round 1

Reviewer 1 Report

In section methods, can you better explain the exclusion criteria?

In section 3, the mechanisms of action of terpenoids could be better described, in particular underlining how the mechanisms could be correlated and useful in the treatment of non- alcoholic fatty liver disease.

I think that table 2 is too rich in information and difficult to read. All the analyzed compounds are reported, with the possible mechanisms of action and the effects from recent studies. You could create several tables, i.e. the first with subclass, compounds, molecular formula, weight, and resources, the second with animal/cell model, dosage, target, mechanism, effect. In the text you could better describe the studies, underling the possible mechanisms and why the mechanisms could be correlated to non-alcoholic fatty liver disease. Are there differences in the doses used in the studies? How are terpenoids administered to animals?

Why don't you describe studies on other carotenoids which are also evaluated in clinical studies (i.e. cryptoxanthyn)?

Reviewer 2 Report

In this review the authors explore the literature of the last 10 years about the use of Terpenoids for NAFLD therapy.

The review is well organized and seems to be quite complete, covering the subject in all aspects. In addition, Table 1 contains all the major information at a glance. 

 Major points

Even if English is not my mother language, I feel that there are serious problems with English language. Reading this work is a bit tiring, due to the construction of many phrases, in a way that sometime it is not easy to understand what the authors mean.  An extensive English editing is mandatory.

Another point is that in this paper it is not easy to understand how many of these compounds have been subjected to a clinical trial and which compounds are actually used in clinics. These issues should be clearly addressed in the Conclusion section, adding a new paragraph.

Round 2

Reviewer 1 Report

Thank you for editing the manuscript as suggested; I have no additional comments, just pay attention to define all the abbreviations. 

Reviewer 2 Report

After the important corrections introduced, the paper can be accepted in the present form.